# Mechanisms of Action of Cassiae Semen for Weight Management: A Computational Molecular Docking Study of Serotonin Receptor 5-HT2C

**DOI:** 10.3390/ijms21041326

**Published:** 2020-02-16

**Authors:** Heidi Yuen, Andrew Hung, Angela Wei Hong Yang, George Binh Lenon

**Affiliations:** 1School of Health and Biomedical Sciences, RMIT University, Bundoora, Victoria 3083, Australia; s3557313@student.rmit.edu.au (H.Y.); angela.yang@rmit.edu.au (A.W.H.Y.); 2School of Science, RMIT University, Melbourne, Victoria 3000, Australia; andrew.hung@rmit.edu.au

**Keywords:** obesity, overweight, weight loss, appetite suppression, herbal medicine, natural product, in silico analysis

## Abstract

Overweight and obesity is a growing global health concern. Current management of obesity includes lifestyle intervention, bariatric surgery and medication. The serotonin receptor, 5-HT2C, is known to mediate satiety, appetite and consumption behaviour. Lorcaserin, an appetite control drug, has demonstrated efficacy in appetite control by targeting 5-HT2C but causes undesirable side effects. This study aimed to explore the potential usage of Cassiae semen (CS), a well-known traditional Chinese medicine used to treat obesity. A computational molecular docking study was performed to determine the binding mechanism of CS compounds to the 5-HT2C receptors in both active, agonist-bound and inactive, antagonist-bound conformations. By comparing binding poses and predicted relative binding affinities towards the active or inactive forms of the receptor, we hypothesise that two of the CS compounds studied may be potent agonists which may mimic the appetite suppression effects of lorcaserin: obtusifoliol and cassiaside B2. Furthermore, two ligands, beta-sitosterol and juglanin, were predicted to bind favourably to 5-HT2C outside of the known agonist binding pocket in the active receptor, suggesting that such ligands may serve as positive allosteric modulators of 5-HT2C receptor function. Overall, this study proposed several CS compounds which may be responsible for exerting anti-obesity effects via appetite suppression by 5-HT2C receptor activation.

## 1. Introduction

Overweight and obesity are growing global health concerns [1]. In 2016, more than 1.9 billion adults aged 18 years and older were overweight, of which over 650 million adults were obese [2]. Weight gain involves excessive fat accumulation in the body, which is an important risk factor for cardiovascular disease (such as coronary heart disease, ischemic stroke and hypertension); type 2 diabetes; fatty liver disease; musculoskeletal disorders like osteoarthritis and certain types of cancers (such as endometrial, breast and colon). These conditions may cause premature death and substantial disability [3,4,5].

Currently, weight management includes three main aspects: lifestyle modification, medication and bariatric surgery [6]. Diet and physical exercise are the main parts of lifestyle modification, but the long-term failure rate of weight-loss programs is very high [7]. Pharmacotherapy is used when lifestyle intervention is insufficient to achieve weight loss or control weight gain. However, most of the anti-obesity medications are associated with undesirable side effects, ranging from mild symptoms of abdominal pain, nausea and vomiting to severe reactions of hypertension, depression, seizures and risk of suicidal behaviour [8]. Some were withdrawn from the market in the past due to their unfavourable risk-to-benefit ratio [9]. Although bariatric surgery is an effective way to lose excess weight, it is usually offered to people who are very obese with a body mass index (BMI) over 40 or people with a BMI over 35 who have other serious health conditions, such as diabetes or heart disease [10].

Due to the limitations of the current weight management regimen, more and more patients are seeking alternative approaches, such as Chinese herbal medicine. Cassiae semen (CS) is a traditional Chinese herb with the function of reducing body weight [11]. A recently published randomised clinical trial (RCT) revealed that a combination of CS and two other Chinese herbs could significantly reduce body weight and BMI compared to a placebo at the end of a 12-week treatment [12]. Moreover, two in vivo studies [13] demonstrated that CS had an anti-obesity effect on rats. 

Although research has shown that CS possesses anti-hyperlipidemia properties [14], the potential effect of CS on appetite suppression has not been investigated. 5-HT2C receptor, a subtype of serotonin receptors, plays an important role in reducing appetite [15,16]. Lorcaserin (LOR), a current anti-obesity drug, is a highly selective 5-HT2C receptor agonist acting as an appetite depressant [17,18,19]. Benzphetamine (BEN) is another anti-obesity drug that suppresses appetite, but its mechanism of action is not fully understood [20,21]. 

The aim of this study is to evaluate the interactions of 5-HT2C receptor to the CS ligands, and hence, to propose the CS compounds responsible for exerting anti-obesity effects via appetite suppression. Computational molecular docking was performed to determine the binding mechanism and predicted binding energies of CS compounds to the 5-HT2C receptor in both active, agonist-bound and inactive, antagonist-bound conformations. By comparing binding poses and predicted relative binding affinities towards the active or inactive forms of the receptor, we proposed a number of ligands which we putatively assigned as potential agonists, antagonists or allosteric modulators. 

## 2. Results and Discussion

### 2.1. CS Compound Interactions with 5-HT2C: Putative Agonists, Antagonists and Allosteric Modulators

Two 5-HT2C serotonin receptor structures were selected as protein targets for the molecular docking studies, as listed in Table 1. These are referred to as “S1” (5-HT2C in an agonist-bound active conformation; chain A of PDBID: 6BQG [22]) and “S2” (5-HT2C in an antagonist-bound inactive conformation; chain A of PDBID: 6BQH [22]). 

We performed docking of 50 CS ligands to S1 and S2 in order to determine the ligands which are most likely to act as agonists, antagonists or as allosteric modulators which bind outside of the canonical agonist-binding pocket. Figure 1 shows the results of the dockings of all 50 ligands, with the ligands binding to a number of clusters in the vicinity of the protein. 

For both S1 and S2, the vast majority of the CS ligands are predicted to bind within the known hydrophobic pocket, as indicated by the large cluster of docked ligands in both Figure 1a,b. This pocket corresponds to the agonist binding site. CS ligands which bind directly to this pocket may potentially act as direct agonists or competitive antagonists. We examine relative differences in the binding affinity at S1 and S2 to assign our studied CS compounds as putative agonists and antagonists as follows.

We hypothesised that ligands which show a more negative binding energy to the agonist-bound 5-HT2C conformation (S1) are more likely to act as agonists; similarly, those that are predicted to exhibit a more negative binding affinity to the known antagonist-bound form of 5-HT2C (S2) are here proposed to serve as antagonists of the receptor.

Table 2 shows the difference in predicted binding affinity values for each CS ligand when docked to S1 subtracted by that of the S2. These differences are described as ∆∆G values in Table 2, where negative values of ∆∆G indicate preferential ligand binding to S2, and thus, hypothesised to act as antagonists, whereas positive values of ∆∆G indicate preferential binding to S1 and are here hypothesised to act as agonists. In addition, ligands which are predicted to bind outside of the agonist binding site may serve as allosteric modulators of 5-HT2C, and these are discussed in subsequent sections. An alternative version of this table, with compounds sorted by S1 binding affinity, is provided in Appendix A (Table A1).

#### 2.1.1. Proposed 5-HT2C Agonists

Inspection of Table 2 indicates that roughly half of all of the 50 CS ligands examined may potentially act as agonists, with positive ∆∆G values. However, most exhibit relatively minor differences. Two compounds exhibit ∆∆G of greater than +1 kcal/mol: c34 (obtusifoliol, PubChem ID 65252), with ∆∆G = +1.3 kcal/mol, and c49 (cassiaside B2 2014, PubChem ID 85185010), with ∆∆G = +1.4 kcal/mol. These are highlighted in blue in Table 2.

Figure 1a shows the predicted most energetically favoured hypothetical binding location of the two putative agonists, c34 and c49, at the agonist-bound conformation of 5-HT2C, indicating, as noted above, that they are predicted to bind to the canonical agonist binding pocket (active site). Furthermore, Figure 2 illustrates the 2D ligand-receptor interaction diagrams for c34 (Figure 2a) and c49 (Figure 2b) bound at the agonist conformation.

For c34, inspection of the contacts indicates that there are 17 residues within 4 Å distance of the ligand, with a combination of nonpolar, polar and acidic residues. Within the active site (Figure 2a), c34 maintained a number of interactions which are similar to those of lorcaserin [15]. There is a close contact with the highly-conserved Asp 134, for which it is known that its involvement is vital to agonist activity [23,24,25,26]. There are also contacts with a number of phenylalanine residues, including Phe 214, Phe 327 and Phe 328, some of which are mediated via π–π ring-stacking interactions with the polycyclic aromatic rings of c34. These ring-stacking interactions are also supported by contacts with a cluster of surrounding nonpolar sidechains within the binding pocket, including Ile 131, Val 135, Val 208 and Leu 209, as well as close contacts with polar residues, including Ser 138, Thr 139, Asn331 and Asn 351. The ligand c34 is also predicted to be sufficiently close to potentially form hydrogen bonding interactions with another acidic residue: Glu 347. This residue is proximal to the hydroxyl group of the ligand c34. The ligand is also predicted to be in close vicinity of Tyr 118, again with the possibility of potentially forming hydrogen bonding interactions with the c34 hydroxyl group. The combination of an additional interaction with Glu 347 and possible hydrogen bonding with Tyr 118 suggests that c34 may be capable of acting as a more potent agonist than LOR and other previously studied agonists. In particular, the roles of Glu 347 in 5-HT2C interactions with ligands is not presently well-characterised and may be worthy of further study.

For the other presently proposed agonist, c49, inspection of the contacts indicates that there are 26 residues within 6 Ang of the ligand, also with a combination of nonpolar, polar and acidic residues. Within the active site (Figure 2b), c49 maintained a number of interactions which are similar to those of c34, described above, as well as those of lorcaserin [15]. These interactions include the key conserved residue Asp 134 and Phe 214, Phe 327 and Phe 328, as well as a cluster of surrounding nonpolar sidechains within the binding pocket, including Ile 131, Val 135, Val 208 and Leu 209. Close contacts with polar residues, including Ser 138, Thr 139, Asn331 and Asn 351, are also present, with Ser 138 also in particular being known to be important to a number of 5-HT2C agonists [27,28]. In addition to the above residues, c34 is also predicted to be sufficiently close to potentially form hydrogen bonding interactions with Tyr 118 and Glu 347 via the hydroxyl group on the ligand’s aromatic ring.

In addition to the above residues, c49 is also predicted to be sufficiently close to potentially form hydrogen bonding interactions with three acidic residues: Asp 134, Glu 347 (also predicted to be present for c34) and Glu 338. The numerous hydroxyl groups available on c49 provides ample opportunity for a hydrogen-bond formation between the ligand and these acidic residues, and c49 may therefore also be a more potent agonist compared to presently known 5-HT2C-active compounds. The compound c49 also forms far more numbers of contacts with other aromatic residues compared to c34, including two tryptophan residues: Trp 130 and Trp 355, which may further indicate the possibility of c49 being an especially potent agonist. The ligand binding roles of the acidic and Trp residues described above have yet to be fully elucidated, and their involvement in the predicted binding mode of putative 5-HT2C agonists is worthy of further investigation.

#### 2.1.2. Proposed 5-HT2C Antagonists

For the proposed antagonists, an inspection of Table 2 indicates that approximately half of the 50 CS ligands examined may potentially act as antagonists, with negative ∆∆G values. As noted above, most exhibit relatively minor differences. However, three compounds which are predicted to bind within the known ligand binding pocket exhibit ∆∆G markedly less than −1 kcal/mol: c06 (2-hydroxymethylanthraquinone, PubChem ID 87014), with ∆∆G = −1.5 kcal/mol; c29 (chrysoobtusin, PubChem ID 155381), with ∆∆G = −2.2 kcal/mol and c47 (physicion-β-d-gentiobioside, PubChem ID 100813), with ∆∆G = −1.5 kcal/mol. These are highlighted in red in Table 2. It is noted that c32 (beta-sitosterol, PubChem ID 222284) exhibits the most negative ∆∆G value of all, with −3.0 kcal/mol, and although this might suggest that it could be hypothesised as a strong antagonist, it is not predicted to bind within the known ligand pocket and instead binds to an external site which we propose may interfere with cholesterol binding. Thus, we excluded c32 from the present discussion regarding possible antagonist.

Figure 1a shows the predicted most energetically-favoured hypothetical binding location of the three putative agonists, c06, c29 and c47, at S1, indicating, as noted above, that they are predicted to bind to the canonical agonist-binding pocket (active site). Furthermore, Figure 3 displays the 2D ligand-receptor interaction diagrams for c06 (Figure 3a), c29 (Figure 3b) and c47 (Figure 3c) bound at the antagonist conformation.

The pattern of interactions for c06 shares many of the same characteristics as those for the putative agonists, described in sections above (Figure 1a). The key characteristics are summarised as follows. Inspection of the contacts indicates that there are 11 residues within 6 Ang of the ligand with a combination of nonpolar, polar, and acidic residues, many of which are similar to those involved in agonist interactions. In particular, nonpolar interactions include π-π ring-stacking interactions with Phe 327, Phe 328 and Trp 324, all of which also exist in agonist binding. Unlike the agonists, there is additional involvement of Phe 223 and Phe 320. These ring-stacking interactions are further supplemented by contacts with a cluster of other nonpolar residues, including Val 135, Ile 142, Leu 209 and Ala 222. Hydrogen-bonding is predicted to form between the hydroxyl group of c06 with Ser 138, and Thr 139 is also sufficiently close to potentially form a hydrogen bond with the ligand.

For c29 (Figure 3b), there are likewise 11 residues within 6 Ang of the ligand. Interactions are formed with several aromatic residues. There is a ring-stacking interaction with Phe 327, and a hydrogen bond mediated via the ligand’s hydroxyl group to Tyr 358. This is again supported by a cluster of nonpolar residues: Val 208, Leu 209, Leu 350 and Val 354. Perhaps, most importantly, in addition to the interaction with the known key residue Asp 134, there is also a close contact formed with a second acidic residue, Glu 347. These residues are also predicted for the putative agonists described in the section above.

Most interestingly, the third predicted antagonist ligand, c47, forms close contacts with 21 nearby residues within the active site (Figure 3c), but, apart from the typical residues described above, this ligand also presents several novel interactions in addition to all of the other presently proposed agonists and antagonists. Moreover, c47 is predicted to form two hydrogen bonding interactions with Ser 334 and Gln 343. Furthermore, unique to this ligand, it is also predicted to be in close vicinity of the basic residue Lys 344. The role of this residue in the ligand binding properties of 5-HT2C is presently unknown. While much attention has focused on the role of the acidic residue Asp 134 in 5-HT2C ligand binding, the impact of the presence of basic residues, such as Lys 344, near the active site is worth further experimental characterisation.

#### 2.1.3. Hypothetical Positive Allosteric Modulators of 5-HT2C

A small number of CS compounds are predicted to bind favourably to 5-HT2C outside of the known agonist binding pocket, all of them being present in the S1 (agonist-bound) form of the receptor with none of the ligands being predicted to bind outside the agonist site for the S2 (antagonist-bound) form of 5-HT2C. These results may suggest that such ligands may serve as allosteric modulators of the 5-HT2C receptor function, and the fact that such novel binding locations are predicted only for the agonist-bound conformation of the receptor may suggest that these compounds act as positive allosteric modulators. Referring to Figure 1a for S1, c32 (beta-sitosterol, PubChem ID 222284) binds to the external surface of 5-HT2C at helical regions that would normally face the hydrophobic tails of the lipid bilayer in which the transmembrane region of the receptor is embedded. In particular, c32 is located within a region corresponding to the upper leaflet of the bilayer. Furthermore, another ligand, c33 (juglanin, PubChem ID 5318717), is also predicted to bind at an external site in a region corresponding to the lower leaflet of the bilayer. These predicted binding locations may suggest that these particular ligands could partition into a bilayer and subsequently interact with the receptor via these external sites.

Many G-protein-coupled receptors exhibit significant responses to membrane cholesterol with regard to the ligand-binding affinity and functional properties, including the 5-HT receptor family. Both locations described above, which we predict to bind CS compounds, suggest the possibility that both beta-sitosterol and juglanin may compete with cholesterol binding sites within the membrane. Previous work has shown the requirement of membrane cholesterol in the organization, dynamics and function of the 5-HT1A receptor [29] and that membrane cholesterol binds preferentially to specific sites on the serotonin 1A receptor. A highly conserved cholesterol recognition/interaction amino acid consensus (CRAC) motif on transmembrane helix V was previously identified as one of the sites with high cholesterol binding capacity, lending support to the notion that it plays a role in the binding of membrane cholesterol. Similarly, another recent computational simulation study [30] identified cholesterol interaction sites in the 5-HT1B and 5-HT2B receptors, with elevated cholesterol density predicted to lie near transmembrane helix 4. Interestingly, in that study, interactions of the endogenous agonist, serotonin with the receptor, is influenced by cholesterol binding at transmembrane helix 4, further highlighting the capability of membrane cholesterol to affect receptor function.

Although cholesterol binding has only been studied in detail, thus far, for the 1A, 1B and 2B subtypes of the 5HT receptor, our present results indicate possible binding locations for compounds c32 and c33, which are polycyclic compounds resembling cholesterol, and raise the intriguing possibility that 5-HT2C may likewise be modulated via the lipid membrane by compounds from cassia seed, perhaps involving competitive binding of these herbal compounds with cholesterol.

Figure 4 displays the 2D ligand-receptor interaction diagrams for c32 (Figure 4a) and c33 (Figure 4b) bound at the agonist conformation. For c32, inspection of the contacts (Figure 4a) indicates that there are seven residues within 6 Ang of the ligand, and the interaction is largely mediated via contacts with nonpolar residues within the hydrophobic transmembrane region of 5-HT2C. These residues include ring-stacking contacts with Phe 220, in concert with other nonpolar residues: Leu 216, Leu 332 and Leu 336. It is noted that the hydroxyl group of c32 is predicted to form no close contacts with any of the residues, suggesting the possibility that the ligand may be aligned such that the polar hydroxyl group protrudes outwards, towards the extramembrane region. As noted in the above sections, c32 binds in a region on 5-HT2C close to those predicted to bind cholesterol in other 5HT receptor subtypes, and it is possible that this ligand could serve to displace cholesterol.

For the other presently proposed positive allosteric modular, c33 (Figure 4b), inspection of the contacts indicates that there are 14 residues within 6 Ang of the ligand with a combination of nonpolar, polar and acidic residues. This ligand contains ample hydroxyl groups, and unsurprisingly, given that its binding position is near a region on 5-HT2C that should lie near the lipid-water interface, it is predicted to form contacts with several polar residues, including Thr 88, Asn 89, Asn 306 and Asn 372. It is also predicted to be close to both an acidic residue, Asp 151, and a basic residue, Arg 152. Its predicted close interactions, in particular with the basic residue Arg 152 close to the lipid-water interface, may suggest that c33 could competitively bind or displace anionic lipids (such as phosphotidylserines or phosphatidylinositols) that normally bind to 5-HT2C.

Both c32 and c33, then, are proposed to affect the 5-HT2C function. Both bind favourably to the predicted positions on the exterior of 5-HT2C only for the agonist-bound conformation of the receptor and are therefore proposed to be positive allosteric modulators. While c32 may also displace cholesterol, the location of c33, closer to the lipid-water interface at the inner leaflet region of the membrane, suggests it may displace anionic lipids within the bilayer in the vicinity of the 5-HT2C receptor.

### 2.2. Common Residues Interacting with CS Compounds

Table 3 shows the residues which form the most common contacts with the proposed ligands of importance in this study. In Table 3, a “1” marks the existence of a contact between the residue (rows) and the ligand (columns) if any atom of the residue and ligand lies within 4 Angstroms of each other. The right-most column sums up all of these interactions for each residue and provides the total number of ligands which make contact with each respective residue. This table illustrates several commonly-interacting residues amongst the CS compounds of interest proposed in this study.

First and foremost, the interaction of the highly conserved residue Asp 134 with molecules is considered to be the most important in enabling agonist activity [15,25]. Other residues that are predicted to be commonly bound by the CS compounds include Val 135, Ser 138, Thr 139, Phe 327 and Phe 328, all of which are well-established as important residues involved in facilitating ligand binding activity for a range of small-molecule organic compounds [28]. For both the predicted agonists and antagonists, interactions typically involve a combination of ring-stacking formations with aromatic residues, supported by clusters of other smaller nonpolar residues, polar residues and acidic residues, including Asp 134.

In addition to the residues of interest known to be important for forming contacts with other ligands in previous studies, this present work has also identified a number of novel residues that include Glu 247, which is predicted to form a second acidic residue supporting anchoring via the classical Asp 134 residue, present in the interaction diagrams for several predicted agonists and antagonists (Section 2.1.1 and Section 2.1.2). Additionally, of interest is the role of basic residues in the active site in terms of ligand binding remains to be elucidated. Lys 344 is one such residue predicted to potentially form hydrogen bonding interactions with a putative antagonist, compound c47, and we suggest that this residue may be worth examining experimentally for understanding its role in ligand-binding to 5-HT2C.

## 3. Materials and Methods

### 3.1. Identification and Preparation of Protein Targets

The structures of serotonin receptors 5-HT2C were searched from the Research Collaboratory for Structural Bioinformatics Protein Data Bank (RCSB PDB) [31].

The identified receptors were generally in complex form with either agonist or antagonist ligands or some other small ligands. The computer software Visual Molecular Dynamics (VMD; University of Illinois, Champaign, IL, USA) [32] was used to prepare the protein targets by separating the protein from the ligands.

Two 5-HT2C serotonin receptor structures were selected as protein targets for the molecular docking studies, as listed in Table 1. These are referred to as “S1” (5-HT2C in an agonist-bound active conformation; chain A of PDBID: 6BQG) and “S2” (5-HT2C in an antagonist-bound inactive conformation; chain A of PDBID: 6BQH).

### 3.2. Identification and Preparation of Ligands

Known drugs for weight management were searched through the online database MIMS [33] for drugs available in Australia and DRUGBANK (Canadian Institutes of Health Research, Alberta Innovates—Health Solutions, and The Metabolomics Innovation Centre (TMIC), Alberta, Canada) [34] for drugs available in the US, Canada and the E.U. Chemical compounds of CS were identified through textbooks [35,36], the China Pharmacopeia [37], an encyclopaedia [38] and the online database TCMSP [39]. All chemical structures were downloaded from the PubChem database (National Center for Biotechnology Information, National Library of Medicine, Bethesda, MD, USA) [40] as SDF files. The SDF files were translated to PDB format with online translator SMILES (National Cancer Institute, Bethesda, MD, USA) [41].

### 3.3. Molecular Docking and Analysis

Molecular docking was performed using AutoDock Vina version 1.1.2 (The Scripps Research Institute, La Jolla, CA, USA) [42]. The docking Graphical User Interface (GUI) frontend PyRx version 0.8 (The Scripps Research Institute, La Jolla, CA, USA) [43] was used to prepare all protein and ligand files for docking and for the generation of docking parameter input files. PyRx was employed to convert all protein and ligand PDB files into PDBQT format. Protonation states for titratable sidechains of the protein were based on those assigned using OpenBabel (OpenEye Scientific Software, Santa Fe, NM, USA) at pH 7. Gasteiger charges were applied to protein and ligands. Docking boxes were set using the “maximise” option in PyRx around the protein receptor in order to enable “blind” docking, in which the entire protein surface and accessible interior pockets were made available for potential binding of ligands. All dockings were performed with the default exhaustiveness value of 8. The dockings were semi-rigid, with full torsional flexibility allowed for the ligands, while the protein receptor structures were kept fixed. AutoDock Vina calculations were performed using the Intel Xeon Sandy Bridge 2.6 GHz Broadwell nodes of the “Raijin” high-performance computing cluster housed at the National Computational Infrastructure (NCI).

Predicted binding affinity for each protein-ligand pair and binding poses of the ligands were generated through the molecular docking process. Binding sites and ligand interactions were analysed with software Maestro [44] from Schrodinger (Schrodinger, New York, NY, USA).

## 4. Conclusions

This study critically investigated the potential usage of Cassiae semen (CS), a well-known traditional Chinese medicine used to treat obesity. A computational molecular docking study was performed to determine the binding mechanism of CS compounds, with the docking poses of the novel compounds compared with interactions known to be vital for ligand binding. Docking of 50 CS compounds were performed on two conformations of 5-HT2C—one in an active, agonist-bound form and the other in an inactive, antagonist-bound conformation.

By comparing binding poses and predicted relative binding affinities towards the active or inactive forms of the 5-HT2C receptor, amongst the ligands predicted to bind to the active site, we hypothesised that two of the cassia compounds studied may be potent agonists: c34 (obtusifoliol) and c49 (cassiaside B2 2014), while three were proposed to be potent antagonists: c06, c29 and c47. Combining the residue interaction results, we have also identified several residues which are commonly bound by the 50 CS compounds.

Several ligands were predicted to bind outside of the agonist binding pocket. These lie on the external surface, roughly in positions consistent with the upper and lower leaflets of bilayer regions. These compounds also bind in roughly the same location as cholesterol for other 5HT receptor subtypes, and we propose that they may competitively bind with cholesterol. Both external-binding ligands also only bind to such sites for the agonist-bound conformation, and as such, it is proposed that they may act as positive allosteric modulators. For these putative allosteric modulator compounds, c32 is proposed to act via the displacement of cholesterol, while c33 is located nearer the lipid-water interface, in contact with acidic and basic residues of 5-HT2C, and may serve to interfere with anionic lipid binding at these locations.

Further work may involve evaluation of the stability of the binding poses predicted in this work using all-atom molecular dynamics simulations to incorporate the effects of protein and ligand movement. The influences of the bilayer in interactions between the proposed allosteric modulators and 5-HT2C in this docking study may also be taken into account using molecular dynamics simulations.

## Figures and Tables

**Figure 1 ijms-21-01326-f001:**
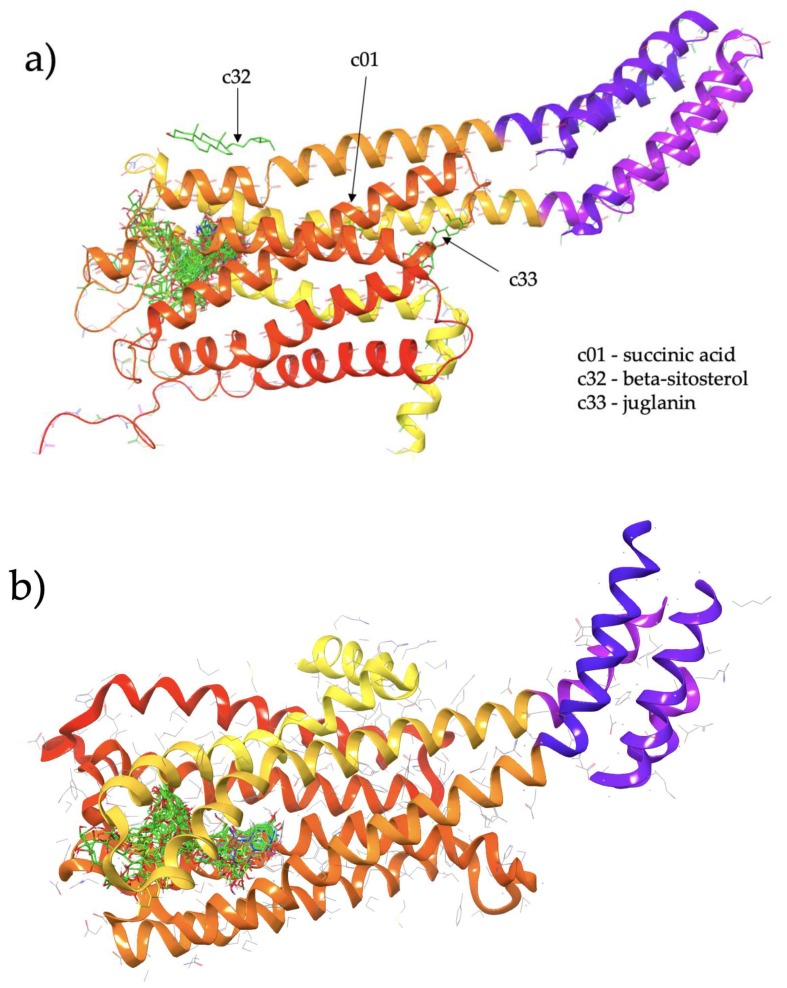
(**a**) Cluster of Cassiae semen ligands binding to structure S1 (5-HT2C in an agonist-bound active conformation; chain A of PDBID: 6BQG). (**b**) Cluster of Cassiae semen ligands binding to structure S2 (5-HT2C in an antagonist-bound inactive conformation; chain A of PDBID: 6BQH).

**Figure 2 ijms-21-01326-f002:**
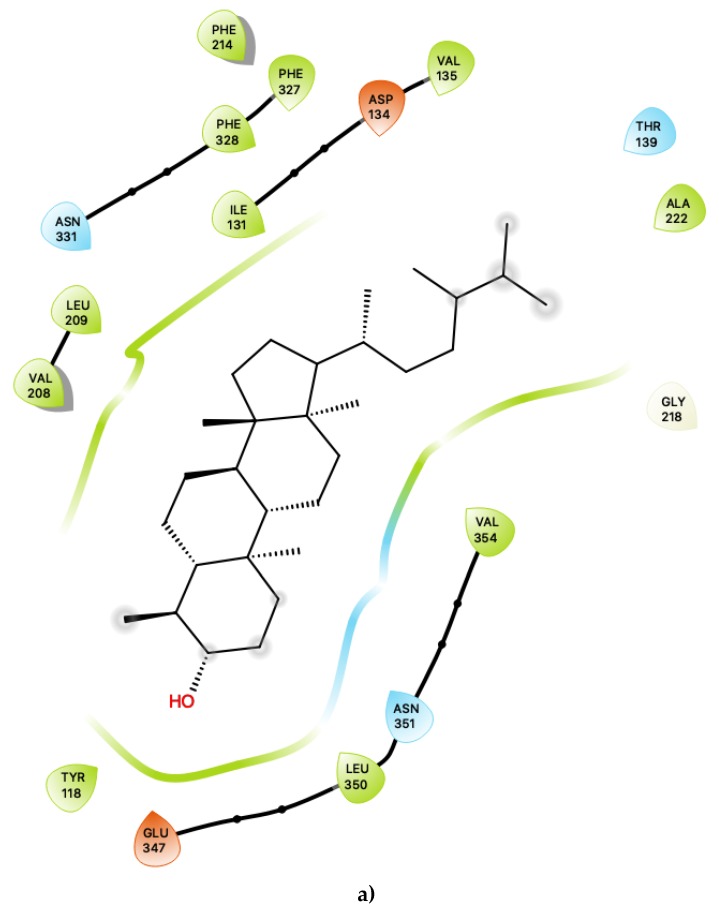
(**a**) Ligand-protein interaction of c34-S1 and (**b**) c49-S1. Arrow indicates hydrogen bond. Grey halos indicate solvent-exposed atoms.

**Figure 3 ijms-21-01326-f003:**
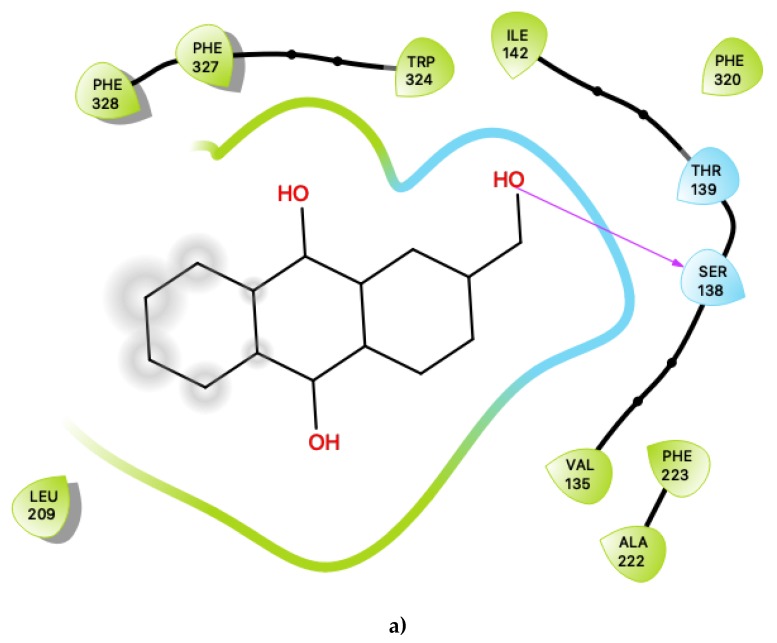
(**a**) Ligand-protein interaction of c06-S2, (**b**) c29-S2 and (**c**) c47-S2. Arrows indicate hydrogen bonds. Grey halos indicate solvent-exposed atoms.

**Figure 4 ijms-21-01326-f004:**
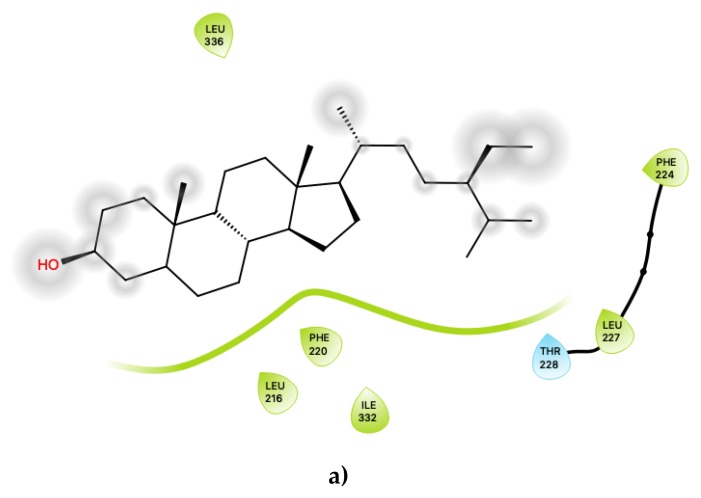
(**a**) Ligand-protein interaction of c32-S1 and (**b**) c33-S1. Grey halos indicate solvent-expose atoms.

**Table 1 ijms-21-01326-t001:** Selected protein targets.

Protein Target	Description
S1	5-HT2C in an agonist-bound active conformation; chain A of PDBID: 6BQG
S2	5-HT2C in an antagonist-bound inactive conformation; chain A of PDBID: 6BQH

**Table 2 ijms-21-01326-t002:** Binding affinity and difference in ∆∆G values between the predicted binding affinity of each Cassiae semen ligand to S1 and S2. Cells for ∆∆G values are colored using a red-white-blue scheme, with blue indicating most positive values and red most negative. Bolded ligand names are ligands of especial interest, discussed in the text.

Ligand ID	Ligand Name	Molecular Weight (g/mol)	Binding Affinity		∆∆G
			(kcal/mol)		(kcal/mol)
			S1	S2	(S2−S1)
c01	succinic acid	118.09	−4.7	−4.2	0.5
c02	2,5-dimethoxybenzoquinone	168.15	−4.9	−5.6	−0.7
c03	5,7-dihydroxychromone	178.15	−7	−7.1	−0.1
c04	(2s,4s)-4-hydroxyarginine	190.24	−5.2	−5.2	0
c05	aspidinol	224.28	−6.1	−6.9	−0.8
**c06**	**2-hydroxymethylanthraquinone**	**238.24**	**−7.8**	**−9.3**	−1.5
c07	quinizarin	240.22	−9.2	−8.8	0.4
c08	chrysophanol-9-anthrone	240.26	−9.2	−9.8	−0.6
c09	uridine	244.20	−7	−7.1	−0.1
c10	torachrysone	246.26	−7.5	−7.7	−0.2
c11	chrysophanol	254.24	−9.1	−9.5	−0.4
c12	rubrofusarin	254.24	−8.4	−8.7	−0.3
c13	chrysin	254.25	−9	−9.2	−0.2
c14	nor-rubrofusarin	258.23	−8.6	−9.2	−0.6
c15	emodin	270.24	−8.9	−9.5	−0.6
c16	aloe-emodin	270.24	−9	−9.2	−0.2
c17	isotoralactone	272.26	−7.2	−8.2	−1
c18	toralactone	272.27	−8.2	−9	−0.8
c19	rhein	284.22	−9.2	−9.3	−0.1
c20	physcion	284.27	−8.9	−9.4	−0.5
c21	questin	284.27	−8.8	−8.4	0.4
c22	obtusifolin	284.27	−8.6	−8.5	0.1
c23	kaempferol	286.24	−8.6	−8.3	0.3
c24	torosachrysone	288.30	−8.5	−8.2	0.3
c25	quercetin	302.24	−8.5	−8.7	−0.2
c26	cassialactone	304.30	−8.3	−8.2	0.1
c27	aurantio-obtusin	330.29	−7.5	−8.3	−0.8
c28	obtusin	344.32	−8.1	−8.3	−0.2
**c29**	**chrysoobtusin**	358.35	**−6.1**	**−8.3**	−2.2
c30	cassiaside A	404.40	−8.5	−9	−0.5
c31	stigmasterol	412.70	−10.7	−10.6	0.1
c32	beta-sitosterol	414.72	−7.3	−10.3	−3
c33	juglanin	418.35	−6.7	−7.9	−1.2
**c34**	**obtusifoliol**	**426.73**	**−10.5**	**−9.2**	1.3
c35	friedelin	426.73	−10.3	−11.7	−1.4
c36	emodin-8-glucoside	432.38	−9.5	−9.7	−0.2
c37	rubrofusarin-6-glucoside	434.40	−8.9	−9.5	−0.6
c38	triacontan-1-ol	438.83	−6.3	−6.2	0.1
c39	gluco-obtusifolin	446.41	−8.9	−9.1	−0.2
c40	betulinic acid	456.71	−8	−9.3	−1.3
c41	gluco-aurantio-obtusin	492.43	−9	−9.4	−0.4
c42	galactomannan	504.44	−6.4	−7.5	−1.1
c43	gluco-chrysoobtusin	520.49	−7.8	−8.5	−0.7
c44	cassiaside B 2017	566.51	−9.5	−10	−0.5
c45	cassiaside C 2019	566.51	−10.2	−10.5	−0.3
c46	rubrofusarin gentiobioside	596.54	−10.1	−9.9	0.2
**c47**	**physcion-β-d-gentiobioside**	**608.55**	**−8**	**−9.5**	−1.5
c48	cassiaside B2 2006	920.80	−9.2	−8.9	0.3
**c49**	**cassiaside B2 2014**	**920.80**	**−9.9**	**−8.5**	1.4
c50	cassiaside C2	920.80	−9.6	−10	−0.4

**Table 3 ijms-21-01326-t003:** Common ligand-binding residues of selected ligands. 1: the existence of a contact between the residue (rows) and the ligand (columns) if any atom of the residue and ligand lies within 4 Angstroms of each other. Green-colored rows indicate residues of especial interest, discussed in text.

Residue	S1-LOC	S1-c03	S1-c09	S1-c22	S1-c34	S1-c49	S2-BEN	S2-c06	S2-c29	S2-c47	Count
Ser110						1				1	2
Ser118	1				1	1				1	4
Trp130						1				1	2
Ile131					1	1					2
Asp134	1	1	1	1	1	1	1		1	1	9
Val135	1	1	1	1	1	1	1	1		1	9
Ser138		1	1	1			1	1			5
Thr139	1	1	1	1	1		1	1			7
ILE142	1						1	1			3
Val185	1		1								2
Val208					1	1			1	1	4
Leu209	1	1		1	1	1	1	1	1	1	9
Phe214	1		1	1	1						4
Val215	1		1			1	1			1	5
Gly218	1	1	1	1	1						5
Ser219	1	1	1								3
Ala222			1	1	1		1	1			5
Phe223							1	1			2
Trp324			1	1			1	1			4
Phe327	1	1	1	1	1	1	1	1	1	1	10
Phe328	1		1	1	1		1	1			6
Asn331	1	1	1		1	1	1		1	1	8
Ser334						1			1	1	3
Glu343						1				1	2
Glu347					1	1			1	1	4
Leu350					1	1			1	1	4
Asn351					1	1			1	1	4
Val354				1	1	1			1	1	5
Tyr358				1		1			1	1	4

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
