# Peer review of "Mechanisms of Action of Cassiae Semen for Weight Management: A Computational Molecular Docking Study of Serotonin Receptor 5-HT2C"

_ijms, 2020, doi:10.3390/ijms21041326_

Round 1

Reviewer 1 Report

Figures are useless with current resolution
page 7, line 168: "+1.5 kcal/mol" should be "-1.5" (compound 47 has a ∆∆G of -1.5 in table 2)
page 12, line 294: "identifies" should be "identified"

One question that arises to me in that kind of work is the lack of molecular dynamics for validating the poses obtained... I think it woud be nice to check some of them (the best ones) and see what happens with both protein (initially fixed) and the ligand (which has rotamers).
That said, it would be nice to have the following information available:
- version of the PyRx used
- have been pKas previously computed? or it's based on openbabel protonation (-xr -p 7)?
- which set of charges has been applied to both protein and ligands (via openbabel: eem, gasteiger, mmff94, ...)?
- which is the geometrical criteria for stablishing the contacts (table 3)?

I don't know if it would be worthy sorting the different ligands on table 2 by one of the binding energies (say S1), in order
not only to focus on the relative binding between S1 and S2, but to emphasize the absolute binding of S1.

Author Response

Dear Reviewer

Thank you for your comments and suggestions

Comment 1:

Figures are useless with current resolution

Response to comment 1:

All figures in manuscript have been replaced with figures of higher resolution.

Comment 2:

page 7, line 168: "+1.5 kcal/mol" should be "-1.5" (compound 47 has a ∆∆G of -1.5 in table 2)

Response to comment 2:

The manuscript has been updated accordingly.

Comment 3:

page 12, line 294: "identifies" should be "identified"

Response to comment 3:

The manuscript has been updated accordingly.

Comment 4:

One question that arises to me in that kind of work is the lack of molecular dynamics for validating the poses obtained... I think it woud be nice to check some of them (the best ones) and see what happens with both protein (initially fixed) and the ligand (which has rotamers).

Response to comment 4:

Molecular simulations will indeed be useful in further exploring the interactions between CS compounds and 5-HT2C, and is a work in progress. We have added the following to highlight this research direction in Conclusions:  

"Further work may involve evaluation of the stability of the binding poses predicted in this work using all-atom molecular dynamics simulations to incorporate the effects of protein and ligand movement. The influences of the bilayer in interactions between the proposed allosteric modulators in this docking study may also be taken into account using molecular dynamics simulations.

Comment 5
That said, it would be nice to have the following information available:
- version of the PyRx used

Response to comment 5

The following details have been added to the Materials and Methods section as follows:

PyRx version 0.8 was used to prepare protein, ligand and docking parameter files.

Comment 6:
have been pKas previously computed? or it's based on openbabel protonation (-xr -p 7)?

Response to comment 6:

Protonation states for titratable sidechains were based on those assigned using OpenBabel at pH 7.

Comment 7
Which set of charges has been applied to both protein and ligands (via openbabel: eem, gasteiger, mmff94, ...)?

Response to comment 7:

Gasteiger charges were applied to protein and ligands. 

Comment 8:
Which is the geometrical criteria for stablishing the contacts (table 3)?

Response to comment 8:

The following detail has been added to the manuscript: “In Table 3, a "1" marks the existence of a contact between the residue (rows) and the ligand (columns) if any atom of the residue and ligand lies within 4 Angstroms of each other.”

Comment 9

I don't know if it would be worthy sorting the different ligands on table 2 by one of the binding energies (say S1), in order
not only to focus on the relative binding between S1 and S2, but to emphasize the absolute binding of S1.

Response to comment 9:

The focus of the manuscript is on the use of relative predicted binding affinity between S1 and S2, rather than absolute predicted binding affinity. Therefore, we have placed a new table as an Appendix, which is a version of Table 2, but sorted according to absolute binding affinity for S1, as suggested by the reviewer.

Reviewer 2 Report

The manuscript entitled "Mechanisms of action of Cassiae semen for weight
management: A computational molecular docking study of serotonin receptor 5-HT2C" presents an exhaustive molecular docking study of different active molecules (obtusifoliol and cassiaside) binding on the serotonin receptor (5-HT2C). The manuscript is well written and present a interesting methodological discussion and analysis. The aims are clearly formulated and the discussion of the results follows the logical sequence of these aims. The topic of the present study is of great importance, especially because clarify the mechanism of the efficient binding of the Cassiae semen (CS) compounds and reveal their biological activity. 

As the only deficiency of the present work I consider to be the low quality of the figures. All of them needs to be rebuilt considering a much higher resolution.

I consider that the present manuscript is suitable for publishing in the IJMS journal after a minor revision. 

Author Response

Dear Reviewer,

We thank you for your valuable comments and suggestions which is important for improving our manuscript.

Comment:

The manuscript entitled "Mechanisms of action of Cassiae semen for weight
management: A computational molecular docking study of serotonin receptor 5-HT2C" presents an exhaustive molecular docking study of different active molecules (obtusifoliol and cassiaside) binding on the serotonin receptor (5-HT2C). The manuscript is well written and present a interesting methodological discussion and analysis. The aims are clearly formulated and the discussion of the results follows the logical sequence of these aims. The topic of the present study is of great importance, especially because clarify the mechanism of the efficient binding of the Cassiae semen (CS) compounds and reveal their biological activity. 

As the only deficiency of the present work I consider to be the low quality of the figures. All of them needs to be rebuilt considering a much higher resolution.

I consider that the present manuscript is suitable for publishing in the IJMS journal after a minor revision. 

Response to comment:

We have re- built all the figures to a much higher resolution.